# Application of the Robust Fixed Point Iteration Method in Control of the Level of Twin Tanks Liquid

**Hamza Khan [1,2,*], Hazem Issa [1] and József K. Tar [1,3]**

[1]    Doctoral School of Applied Informatics and Applied Mathematics, Óbuda University, 1034 Budapest, Hungary; hazem.issa@phd.uni-obuda.hu (H.I.); tar.jozsef@nik.uni-obuda.hu (J.K.T.)
[2]    Department of Textile & Clothing, National Textile University Karachi Campus, Karachi 74900, Pakistan
[3]    Antal Bejczy Center for Intelligent Robotics (ABC iRob), University Research and Innovation Center, Óbuda University, 1034 Budapest, Hungary
*    Correspondence: hamza.khan@phd.uni-obuda.hu; Tel.: +92-333-3032982

**Abstract:** Precise control of the flow rate of fluids stored in multiple tank systems is an important task in process industries. On this reason coupled tanks are considered popular paradigms in studies because they form strongly nonlinear systems that challenges the controller designers to develop various approaches. In this paper the application of a novel, Fixed Point Iteration (FPI)-based technique is reported to control the fluid level in a "lower tank" that is fed by the egress of an "upper" one. The control signal is the ingress rate at the upper tank. Numerical simulation results obtained by the use of simple sequential Julia code with Euler integration are presented to illustrate the efficiency of this approach.

**Keywords:** robust fixed point transformation; fixed point iteration; dual fluid tanks; control; fpi-based simulations; level control of liquids

## 1. Introduction

Tackling nonlinear problems in Control Engineering (CE) is an emerging field of the modern era, it affects our life at several aspects [1]. Control of level of liquids, in coupled tank systems, has prime importance in process industries. Many industries and production units can be enlisted where the importance of the level of the fluid stored in several tanks play significant role, for instance, in food processing units, dairy filtration, plants of nuclear power generations, companies dealing with pharmaceutical products, in system of water purification units, spray coating industries, etc. Furthermore, controlling the level of liquids in industries is also remarkably useful to enhance the economic beneficial and quality of the products prospects (e.g., [2]). In the practice this field is very rich and several design techniques have been developed in nonlinear control. In [3] Li et al. elaborated a case study on the application methods and explained several points about process control by taking linear and nonlinear tanks as examples, which were further explained in detail by simulations. The chain of coupled tanks form a typical nonlinear system because the egress rate of a tank is nonlinear function of the fluid pressure, i.e., the height of the liquid in the tank. However, if only positive input rate is available as control action, the problem is further burdened by truncation-type nonlinearities. According to [4] mathematical tank models can be obtained by analytical and experimental methods while for the Coupled Fluid Tanks (CFT) system those models can be obtained by applying the laws of energy and mass conservation, and the model describing the behavior of the liquid flow in the exit tap that contains nonlinear terms for turbulent flow.

It is important to know that first time the automatic liquid level control system for a laundry appliance was patented in 1981 [5], and in 1992 the earliest study was reported on controlling the liquid by using capacitive sensors [6]. This invention consisted in a liquid level control system for selectively activating and deactivating a pump according to the liquid level indicated by the capacitive sensors. In [7] the authors elaborated a significant and feasible nonlinear feedback control. To explain this concept it was further demonstrated by a case study of the control performance of water tank level. In several studies on controlling the dual or longer chains of tanks Sliding Mode Control (SMC) was employed by several researchers (e.g., [8–11]). In [8] authors have successfully reduced the chattering problem associated with SMC by proposing two different dynamic schemes. They have presented the results by successful simulation performed in MATLAB. An experimental set-up was illustrated that guaranteed asymptotic stability of the closed loop system. The design and analysis of sliding mode controller by simulation and experimental results were presented in [12] to demonstrate the effectiveness of the proposed controllers proving its capability of dealing with intrinsic uncertainties of the model and its superior performance than the traditional PI control.

Struggles from various aspects using several methods and tuning techniques can be studied in the literature (e.g., by [13,14]). In [15] authors studied the dual fluid tanks level control by comparing the efficiency of the parameter estimation method with the performance of numerical derivation technique. The method was based on the estimation of the ultra-local model parameters instead of using numerical derivation technique. An adaptive PI method was applied in order to overcome the influences of the perturbations and noise output signals. To optimize the PID parameters of dual fluid tanks system controller,a design methodology using Particle Swarm Optimization (PSO), was proposed and it was claimed that this method provides efficient results in comparison to the genetic algorithm-based method in a shorter and better time resolution [1]. To implement an improved neural network-based approximation Dynamic Programming named Action-Depended Dual Heuristic Dynamic Programming (ADDHP) was viably used by omitting the model network completely in [16]. The proposed method successfully provided the results where only the use of the states of the present and previous time steps were considered to calculate the derivatives of the performance function by avoiding the prediction of the states of the next time step.

Maintaining the fluid range in a pre-defined, desired level is a practical paradigm used in several control teaching laboratory experiments [17]. For this purpose the mathematical model of the plant can be used in the control design. Further study of the Comparison of the Operation of Fixed Point Iteration-based Adaptive and Robust VS/SM-type solutions for controlling "Two Coupled Fluid Tanks" (TCFT) have been recently done in [18], (that was accepted for publication), where it was comparatively proved that rather VS/SM type solutions FPI-based solutions are more efficient and applicable.

The fluid level can be controlled by determining the input flow rate in tanks that in the same time produce some outputs. The specified level of liquid in one of the tanks is called the set point and it expresses the process variable. The error can be defined by the difference value of the actual liquid level and the set point. By measuring the process variable the controller decides to set the appropriate position. In this study the level control of fluid in the pre-selected range of the below tank shown in Figure 1 was studied using the novel Fixed Point Iteration (FPI)-based method from a new and particular prospect. The method is successfully applied, and viable results are produced using several adaptive parameters in Julia language by writing a simple programming code.

In the sequel the paper is organized as follows; Section 2 explains the method used in this study and the software which is used to write the code for simulation results. In Section 3 the details of the FPI-based method is discussed briefly. Section 4 describes the case study of the Dual Tank systems and the equation of motions of the system in detail. Control task of the system and the details related to the procedure is

discussed in Section 5. Using simulation the required produced results are discussed with parameters and their values in Section 6. The conclusion of the study is given in Section 7.

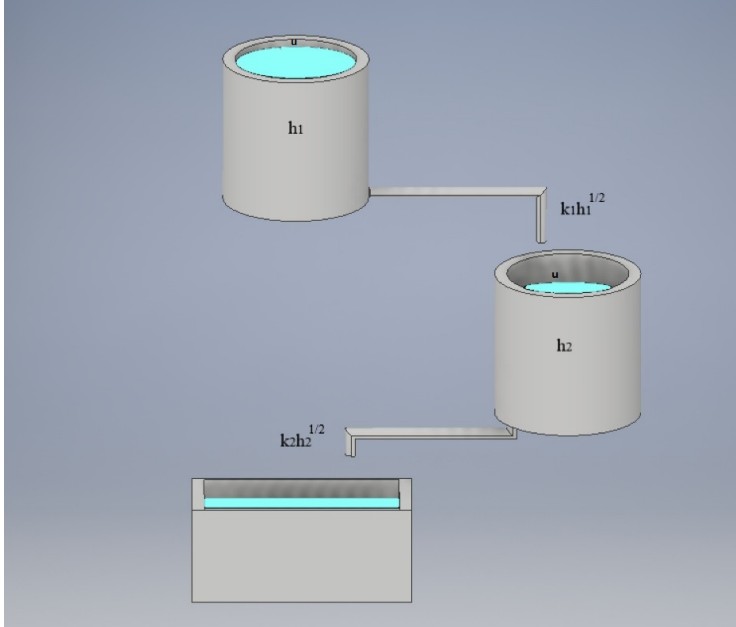

**Figure 1.** Model of the fluid tanks adjusted in up and down positions.

## 2. Methodology

In plant and industrial process system, based on strongly complex nonlinear systems, basic knowledge of mathematical analysis and information about the nature of the controlled system is required. In such complex nonlinear systems, manual study for the derivation of the precise analytical model, is almost impossible. The controller can be designed by the use of some available approximate model (it may have either "analytical form" or "soft computing-based formulation"), and the effects of modeling imprecision can be compensated either by robust or adaptive techniques. The operation of the various approaches normally can be compared by the use of numerical simulations. To conduct this research authors used JULIA language where a simple sequential code was developed to perform the task of level control of the fluid in CFT systems. he JULIA language is one of the fastest high-level and high-performance computer programming tools for dynamical systems, and it can be used for obtaining numerical solutions in computational sciences, which works e.g., on the ATOM based environment using libraries/repositories of other programming languages e.g., MatPlotLib of Python, Fortran etc., The well-known Fixed Point Iteration (FPI) method was used to study the fluid level in the lower tank of vertically fixed dual tanks.

## 3. On the Concept of Using of Fixed Point Iteration-Based Control Method

Based on certain antecedents, it was Stefan Banach who introduced the concept of *linear, normed, complete metric spaces*, that after him, later, were named "Banach Spaces". He systematically used this concept in proving various theorems via constructing contractive functions that map a Banach space into itself. In his "Fixed Point Theorem" [19] in 1922 proved that by the use of such functions iterative Cauchy sequences can be generated that converge to the unique fixed point of these functions. His invention successfully extended the set of classic mathematical methods as e.g., the Newton-Raphson algorithm that is regarded to be one of the most efficient ones even in our days [20].

As an alternative of the Lyapunov function-based adaptive control technique that was based on Lyapunov's PhD dissertation [21], and became known by the Western World in the sixties of the past century [22], Banach's method was introduced in the adaptive dynamic control in [23]. In this approach, Banach's iterative technique was used so that finding the appropriate control signal was transformed to finding the fixed point of a contractive map in an iterative manner for which, during one digital control cycle, only one step of this iteration could be done. At the beginning the available approximate dynamic model is used, and its input is step by step so deformed that finally the response of the controlled system corresponds to the "desired", purely kinetically calculated one. In this early approach the "Robust Fixed Point Transformation (RFPT)" was used for transforming the control task into a fixed point problem. Later on to confirm its applicability several papers were produced using this FPI method from various applicable aspects (e.g., [24–27] etc.) and other functions were introduced too, for this task (e.g., [28–30]). A brief discussion about its applicability and working procedure is given below.

Consider the equation of motion of the controlled system in the differential form $\dot{h} = f(h, u)$, where $h \in \mathbb{R}^M$ is for the state variable of the system, $u \in \mathbb{R}^K$ is called a control signal. The initial condition of the motion can be expressed as $h_0 \equiv h(t_0)$ which is given in advance. Using the inverse of the equation given in Equation (2) the necessary calculated force can be expressed in mathematical form as $u^{Est}(t) = f_{Appr}^{-1}\left(h(t), \dot{h}^{Des}(t)\right)$ that is clearly described in Equation (5). The realized state-drift can be expressed as $\dot{h}(t) = f_{Exact}\left(h(t), u^{Est}(t)\right)$, where the estimated control variable is involved. In this manner the function $\psi$ can be defined as a "response function" that depends on the exact and the inverse approximate models: $\dot{h} = \psi\left(h(t), \dot{h}^{Des}(t)\right)$. Because the available approximate model is not precise, $\dot{h}$ cannot be equal to the desired one as $\psi\left(h(t), \dot{h}^{Des}(t)\right) \neq \dot{h}^{Des}(t)$. In this kind of adaptive control the basic idea is to find a suitably deformed input argument e.g., $r_\star$ for that $\psi$ can be equal to the desired one as $\psi\left(h(t), r_\star(t)\right) = \dot{h}^{Des}(t)$. Hence, this deformed value can be a limit of an iterative sequence $\left\{r_0 = \dot{h}^{Des}(t), \ldots, r_{k+1} = G\left(r_k, \psi\left(x(t), r_k\right), \dot{h}^{Des}(t)\right), \ldots\right\}$, where $G$ is another function that expresses the Robust Fixed Point Transformation:

$$G\left(r, \psi\left(h(t), r\right), \dot{h}^{Des}(t)\right) \overset{def}{=} (r + K)\left[1 + B \tanh\left(A\left(\psi\left(h(t), r\right) - \dot{h}^{Des}(t)\right)\right)\right] - K, \tag{1}$$

where $A$, $B$ and $K$ are used as the adaptive control parameters. In many cases we have obtained efficient and acceptable results by the use of suitably determined adaptive control parameters (e.g., [31,32]). Here, if $|K| \gg |r|$ and a very small value of $A$ are chosen, the iteration converges to the desired point $r_\star$ if the sign of the first derivative of $\psi$ with respect to $r$ i.e., $\frac{d\psi}{dr}$ does not fluctuate. It is evident by inserting $r_\star$ into Equation (1) that $G\left(r_\star, \psi\left(h(t), r_\star\right), \dot{h}^{Des}(t)\right) = r_\star$, that means that $r_\star$ is the fixed point of function $G$. The contractivity must be guaranteed for the convergence of the function by suitably fixing the values of the adaptive parameters.

## 4. Study of Dual Tank Systems

The two-tank-system under consideration is described in Figure 1 where the liquid tanks are fixed on upper and lower position and the liquid spills from upper i.e., tank 1 to the lower i.e., tank 2. The liquid level is predefined for tank 2 as function of the time. The two identical liquid tanks have the same section expressed by S m$^2$. The water level in the upper tank is denoted by $h_1(t)$, which is the first state variable. Similarly, the water level in the lower tank 2 is expressed by $h_2(t)$ (i.e., the second state variable). Suppose that a pump can be adjusted that pumps the liquid into the first (upper) tank by a connected pipe. Variable $q_1(t)$ m$^3 \cdot$ s$^{-1}$ expresses the input flow of the upper tank after pumping by the pump, $q_2(t)$ m$^3 \cdot$ s$^{-1}$ is the output flow of the upper tank and $q_3(t)$ m$^3 \cdot$ s$^{-1}$ denotes the output flow of the lower tank. In the steady state, the conservation of the total volume of water leads to $q_1(t) = q_3(t)$.

*Model Representation*

The exact nonlinear models of the considered system are as follows:

$$\dot{h}_1 = -\frac{k_1\sqrt{h_1} - u}{S}; \dot{h}_2 = \frac{k_1\sqrt{h_1} - k_2\sqrt{h_2}}{S} \tag{2}$$

in which $k_1$, $k_2$ $\left[\text{m}^{5/2} \cdot \text{s}^{-1}\right]$, and $S$ $\left[\text{m}^2\right]$ are parameters of the outlet assuming turbulent egress.

## 5. Control Task of the Dual Tank System

Our task is to control $h_2$ by using the control signal $u$. It can be observed in (2) that the first derivative of the controlled variable $h_2$ i.e., $\dot{h}_2$ does not depend directly on the control signal. To establish a relationship between them, the 2nd time-derivative of $h_2$ is used which directly depends on the 1st time-derivative $\dot{h}_1$ of $h_1$. According to (2) this directly depends on the control signal. By this relation it can be seen that the relative order of the control task is 2. By using the chain rule of differentiation it is obtained that

$$\ddot{h}_2 = \frac{k_1}{2S}h_1^{-1/2}\dot{h}_1 - \frac{k_2}{2S}h_2^{-1/2}\dot{h}_2 \ . \tag{3}$$

From (2) $\dot{h}_1$ can be substituted into (3) that results in

$$\ddot{h}_2 = \frac{k_1 h_1^{-1/2}}{2S^2}u - \frac{k_1^2}{2S^2} - \frac{k_2}{2S}h_2^{-1/2}\dot{h}_2 \ . \tag{4}$$

Therefore for a "desired 2nd time-derivative of $h_2$", $\ddot{h}_2^{Des}$, the model using the available approximate parameters is:

$$u = \frac{2S^2 h_1^{1/2}}{k_1}\ddot{h}_2^{Des} + k_1 h_1^{1/2} + \frac{k_2 S h_1^{1/2}\dot{h}_2}{k_1 h_2^{1/2}} \ . \tag{5}$$

Equation (5) must be completed with the restriction that the feasible control signal is $u \geq 0$. If this equation requires negative control signal, it is truncated at zero.

By the comparison of adaptive FPI-based and Robust VS/SM-type control solutions by [18] it was concluded that adaptive FPI-based solutions are more feasible and efficient. For the system's desired second time-derivative in principle a great variety of functions can be so invented in the form $\ddot{q}^{Des} = f(\ddot{q}^N, \dot{q}^N, q^N, \dot{q}, q, t)$ that $\|q^N(t) - q(t)\| \to 0$ as $t \to \infty$. In the simple PD type controller for the control of $n \in \mathbb{N}$ order systems where the $n^{th}$ order time-derivative of $q$ can be instantaneously adjusted by the control action. To specify and establish the "desired trajectory tracking error relaxation" the trajectory tracking error can be introduced as the difference of the *"Nominal Trajectory"* to be tracked ($q^N(t)$), and the realized one ($q(t)$) as

$$e(t) \overset{def}{=} q^N(t) - q(t) \tag{6}$$

with a constant parameter $\Lambda > 0$ *a purely kinetically defined PD-type tracking policy* can be chosen that aims at the realization of the error relaxation according to

$$\ddot{q}^{Des} = \ddot{q}^N + \Lambda^2 e + 2\Lambda\dot{e} \ . \tag{7}$$

In the paper [18] it was found that too big $\Lambda$ may cause instabilities at the beginning when the initial error is large, however, later, for more precise and fast tracking a greater $\Lambda$ value would be expedient. If this value is not exactly constant but it slowly varies in comparison with the "dynamics" of the signal to

be tracked, its slow variation does not mean significant problem in the error relaxation. For this purpose, in this paper its value was set according to the rule

$$\Lambda_1 = \Lambda_{max}\sqrt{\frac{e_{max}}{e_{max} + |e|}}; \Lambda_2 = \frac{\Lambda_{max}\dot{e}_{max}}{\dot{e}_{max} + |\dot{e}|}; \Lambda = \min(\Lambda_1, \Lambda_2)\ , \tag{8}$$

in this manner great tracking and tracking velocity errors set relatively small feedback, while for small errors the maximal value $\Lambda_{max}$ is well approached.

For the adaptive deformation of the input of the available approximate system model the classic RFPT function given in Equation (1) was used. It has to be noted that at the beginning some initial information is needed on the operation of the non-adaptive controller. Abrupt switching on the adaptivity may cause a local shock that can be avoided by introducing the adaptive deformation with a continuously increasing weight function of time in the following manner:

$$w(t) = 0 \text{ for } t < t_{ad}\ , \tag{9}$$

$$w(t) = \tanh\left(\frac{t - t_{ad}}{T}\right) \text{ for } t \geq t_{ad}\ , \tag{10}$$

$$\ddot{q}^{Def}(t_n) = w(t_n)G\left(\ddot{q}^{Def}(t_{n-1}), \ddot{q}^{Des}(t_n), \ddot{q}(t_{n-1})\right) + (1 - w(t_n))\ddot{q}^{Des}(t_n)\ , \tag{11}$$

in which the adaptive deformation is switched on at time $t_{ad}$, the parameter $T$ s determines the typical time for reaching the full deformation, $t_n$ belongs to the commencement of the $n$th discrete control cycle. (When the control signal is truncated at $u = 0$ the adaptivity is switched off, and later, when its reactivation becomes possible, it is turned on again, gradually.)

## 6. Simulation Results for the Coupled Tank

In the simulations the nominal trajectory to be tracked was a "chirp signal" that corresponds to a "sinusoidal" function with slowly increasing frequency. The model and control parameters are given in Table 1, the time-resolution of the Euler integration was $\Delta t = 10^{-2}$ s. In the first simulations in Equations (9)–(11) the $T \ll \Delta t$ value corresponded to "abrupt switching on" of the adaptivity when the control signal $u$ left the negative, truncated region.

For setting the value of $\Lambda_{max}$ numerical simulations were done. It was found that for $\Lambda_{max} \geq 6\,\text{s}^{-1}$ the results were distorted, that means a kind of speed limitation of the control. Gradual increase of the positive value of $\Lambda_{max}$ from 1 improved the trajectory and the desired trajectory went closer to the nominal value. Typical results are given in Figure 2 in which typical "overshooting" are observable for large values.

**Table 1.** The parameters used in the simulations.

| Parameter | Exact Value | Approximate Value |
|:---:|:---:|:---:|
| $S$ | $1.0 \text{ m}^2$ (model parameter) | $1.8 \text{ m}^2$ |
| $k_1$ | $0.02 \text{ m}^{5/2} \cdot \text{s}^{-1}$ (model parameter) | $0.005 \text{ m}^{5/2} \cdot \text{s}^{-1}$ |
| $k_2$ | $0.03 \text{ m}^{5/2} \cdot \text{s}^{-1}$ (model parameter) | $0.0058 \text{ m}^{5/2} \cdot \text{s}^{-1}$ |
| $h_{1_0}$ | $3.0 \text{ m}$ (initial height) | Not Applicable |
| $h_{2_0}$ | $0.5 \text{ m}$ (initial height) | Not Applicable |
| $\Lambda_{max}$ | $5.5 \text{ s}^{-1}$ (control parameter) | Not Applicable |
| $K$ | $40 \text{ m} \cdot \text{s}^{-2}$ (control parameter) | Not Applicable |
| $B$ | $-1$ (control parameter) | Not Applicable |
| $A$ | $\frac{10^{-1}}{K}$ (control parameter) | Not Applicable |
| $e_{max}$ | $0.1 \text{ m}$ (control parameter) | Not Applicable |
| $\dot{e}_{max}$ | $0.1 \text{ m} \cdot \text{s}^{-1}$ (control parameter) | Not Applicable |

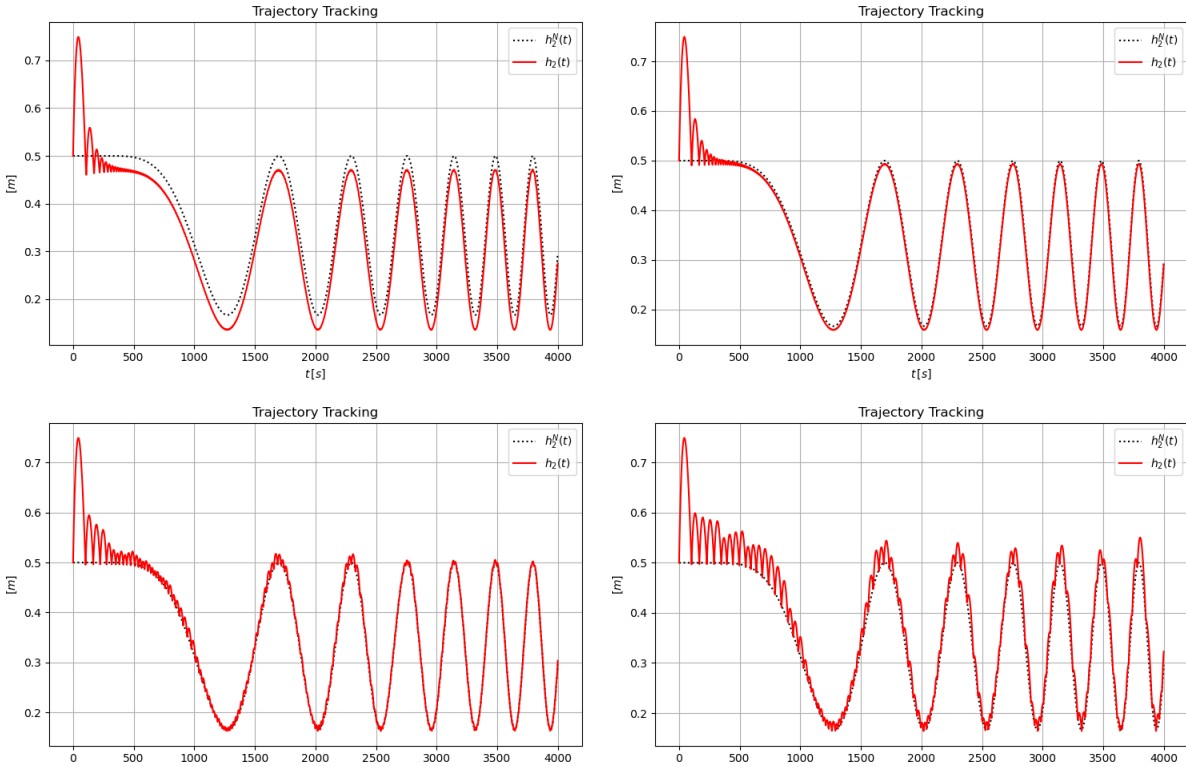

**Figure 2.** Trajectory tracking for ($\Lambda_{max} = 1$ (upper Left Hand Side), 4 (upper Right Hand Side), 9 (bottom Left Hand Side), 12 (bottom Right Hand Side)).

A quite improved and viable result was observed with $\Lambda_{max} = 5.5$ shown in Figure 3. The zoomed in excerpts show that the controlled variable $h_2$ well approached the nominal one $h_2^N$. The tracking error showed a drastic jump in the beginning and then went closer to zero, too.

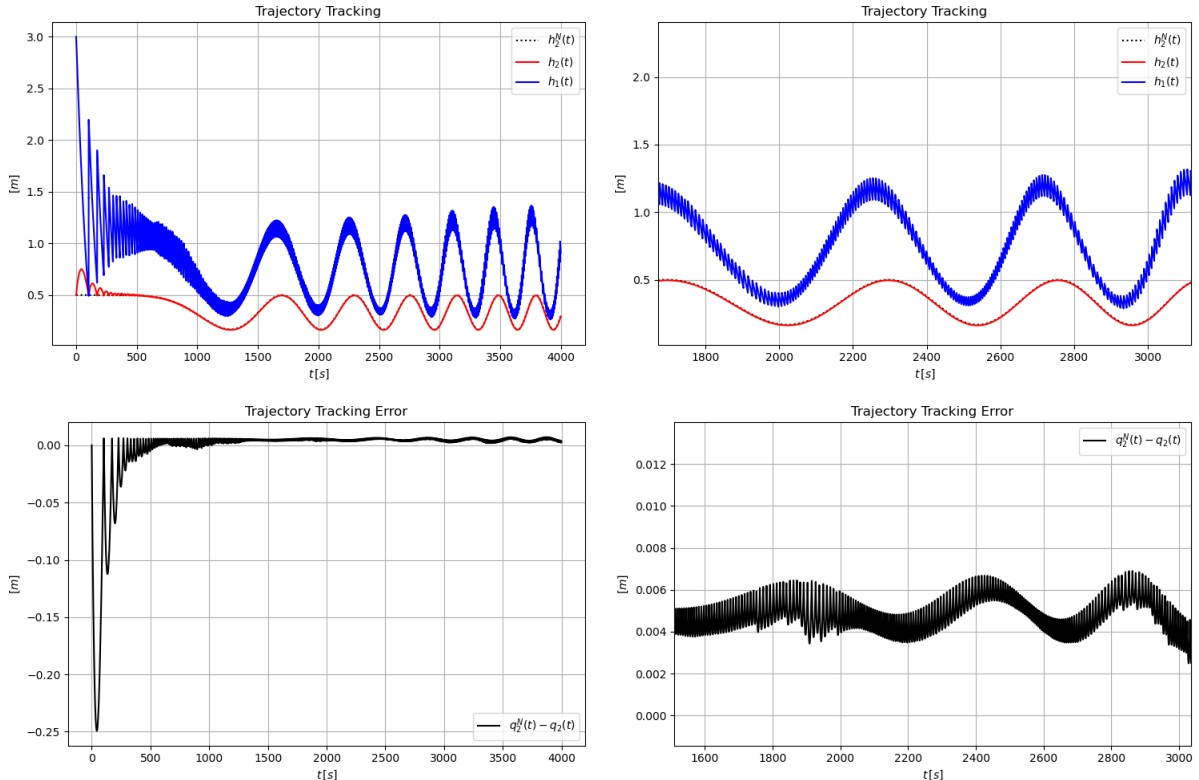

**Figure 3.** For $\Lambda_{max} = 5.5$ with adaptivity: Trajectory tracking with its zoomed in excerpt (upper Left & Right Sides) and tracking errors with its zoomed in excerpt (bottom Left & Right Sides).

In Figure 4 the control signal $u$ and the second time-derivatives of $h_2$ are shown with their zoomed in excerpts. It is evident that the control signal was frequently truncated at 0 when no adaptivity was applicable.

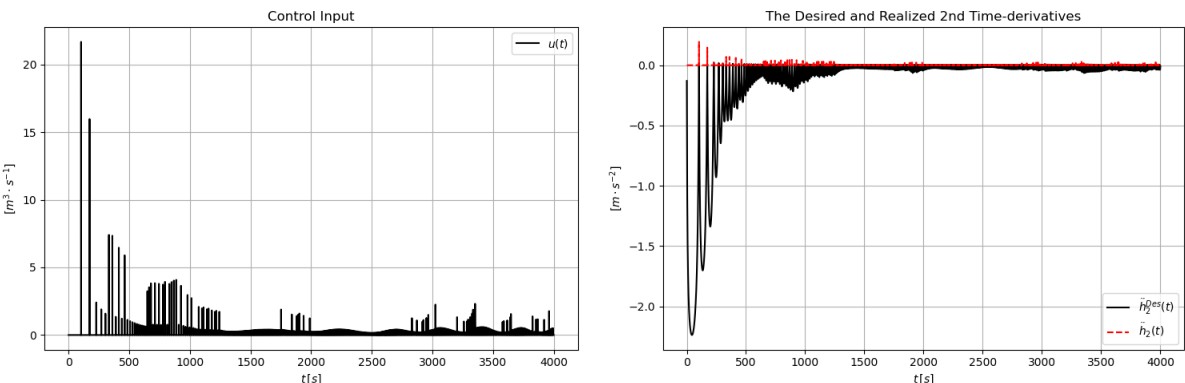

**Figure 4.** The adaptive control input and the second time-derivatives with their zoomed in excerpts for $\Lambda_{max} = 5.5$ with adaptivity.

When the adaptivity was switched off during the whole simulation for the same parameter settings, considerable errors are revealed by Figures 5 and 6.

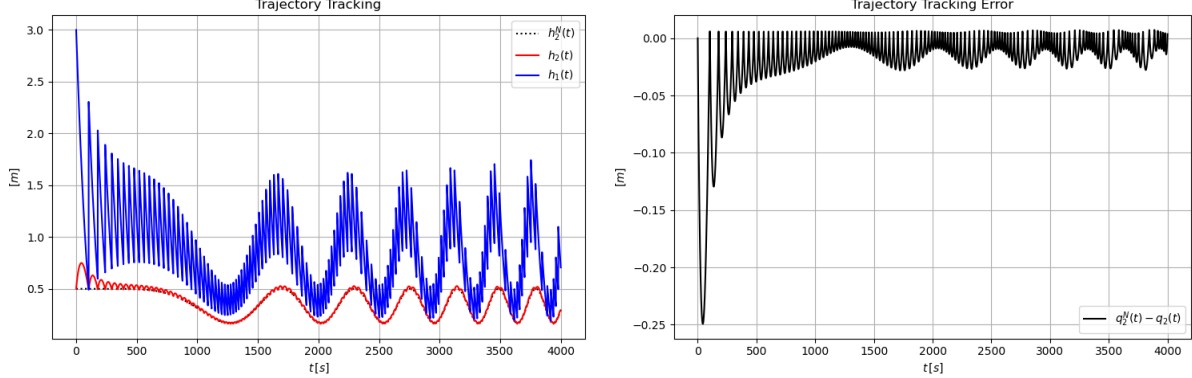

**Figure 5.** Trajectory tracking and tracking errors without adaptivity.

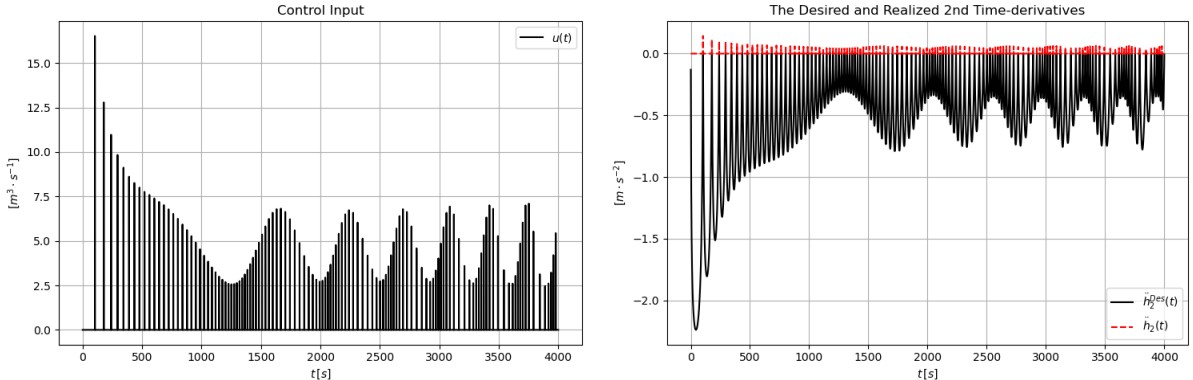

**Figure 6.** The Control input and second time-derivatives without adaptivity.

It is worthy of investigation how the gradual switching on the adaptivity concerns the control. In the following investigations the $T = 5\Delta t$ value was set that means that the adaptivity was gradually turned on during approximately 5–7 digital steps. Figures 7 and 8 reveal considerable reduction in overshooting and the appearance of a smooth signal in spite of the drastic modeling errors in the system's parameters.

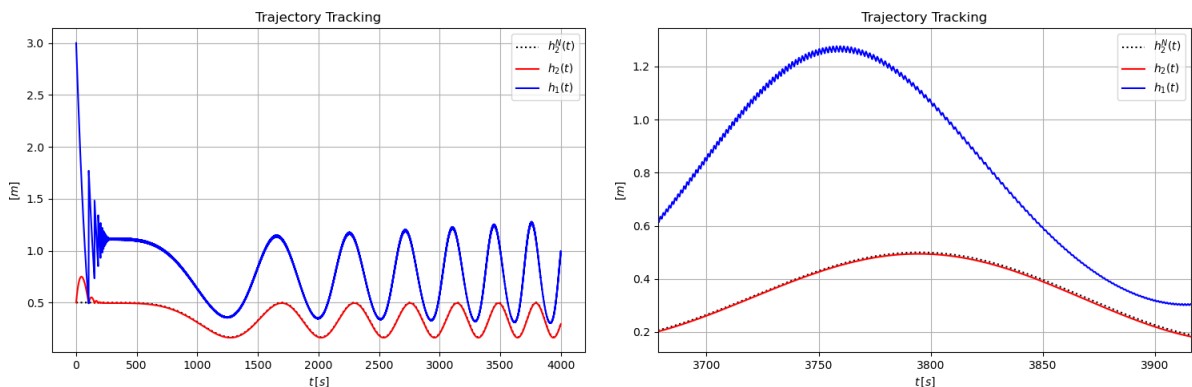

**Figure 7.** *Cont.*

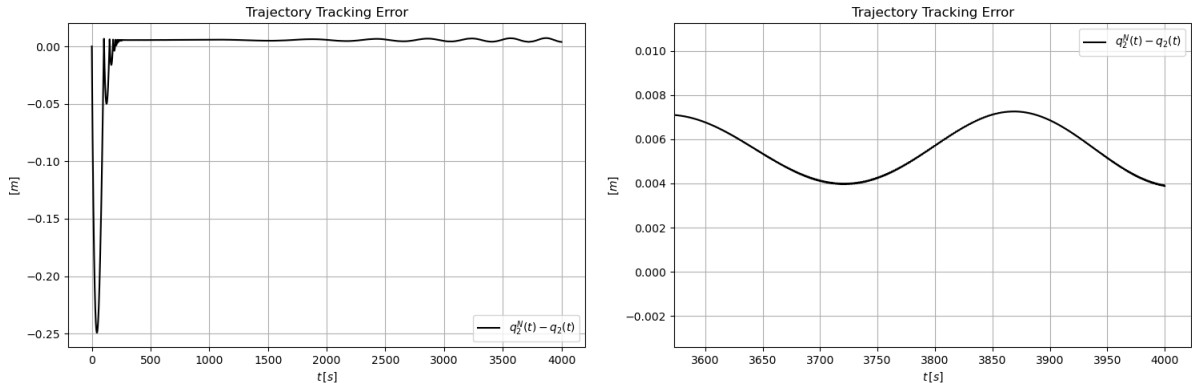

**Figure 7.** For $\Lambda_{max} = 5.5$ and $T = 5\Delta t$ with adaptivity: Trajectory tracking with its zoomed in excerpt (upper Left & Right Sides) and tracking errors with its zoomed in excerpt (below Left & Right Sides).

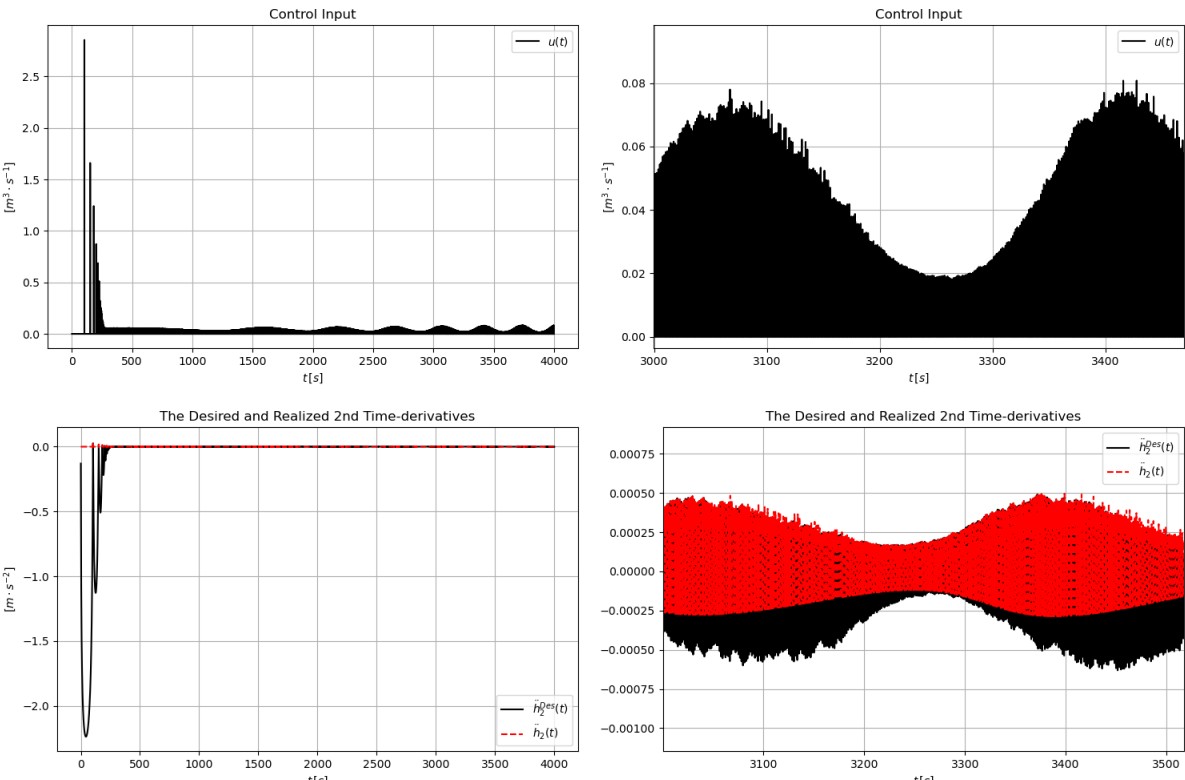

**Figure 8.** For $\Lambda_{max} = 5.5$ and $T = 5\Delta t$ with adaptivity: The adaptive control input with its zoomed in excerpt (upper Left & Right Sides) and the second time-derivatives with its zoomed in excerpt (below Left & Right Sides).

## 7. Concluding Remarks

In this paper the operation of an improved, fixed point iteration-based adaptive controller's operation is investigated via numerical simulations. This approach is a mathematically quite simple alternative of the traditional, Lyapunov function-based adaptive control design in which the *kinematic* and *dynamic aspects* of the desired control operation are sharply separated from each other. The tracking requirements (normally the relaxation of the trajectory tracking error) is formulated at first, then, by the use of the

available approximate system model, the input of this model is deformed in an adaptive manner until the realization of the kinematic requirements is achieved. The method is based on Stefan Banach's fixed point theorem from 1922, which is so utilized in a digital control, that during one control step, one step of the adaptive iteration can be executed.. The improvement of this approach consisted in using tracking error-dependent kinematic error feedback parameters and gradually switching on the adaptation rule, so avoiding an "initial shock".

For testing the method the popular paradigm of two coupled liquid tanks was considered in which the ingress fluid rate of the first tank was used for controlling the fluid level in the second tank. Besides being a strongly nonlinear, relative order 2 control task, the problem was made more complicated by the physical impossibility of applying "negative control signal". Whenever fulfilling the kinematic requirement required negative control action, the control signal was truncated at zero, and the adaptivity was switched off until the appearance realizable positive control signals that made it possible to observe the effects of the adaptive deformation again. This simple physical system can serve as a paradigm for chemical systems' control and control issues in life sciences (e.g., treating the illness type 1 diabetes mellitus) in which the control input corresponds to the ingress of some pure reagent as e.g., insulin, but a pure reagent cannot be extracted from the system, i.e., no negative control signal can be applied. In further research the noise sensitivity of the method seems to be worthy for investigation.

**Author Contributions:** Conceptualization, J.K.T. and H.K.; methodology, H.K.; software, ; H.K., H.I. and J.K.T.; validation, H.K., H.I. and J.K.T.; formal analysis, H.K.; investigation, H.K., H.I. and J.K.T.; writing—original draft preparation, H.K.; writing—review and editing, H.K., H.I. and J.K.T.; supervision, J.K.T.; Resources, J.K.T.; funding acquisition J.K.T. All authors have read and agreed to the published version of the manuscript.

**Funding:** Óbuda University, H-1034 Budapest, Bécsi út 96/B Hungary has provided funds for this research.

**Acknowledgments:** Authors are highly acknowledge the support of the Doctoral School of Applied Informatics and Applied Mathematics, Óbuda University Budapest, Hungary.

**Conflicts of Interest:** The authors declare that they have no conflict of interest.

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
