# Peer review of "Application of the Robust Fixed Point Iteration Method in Control of the Level of Twin Tanks Liquid"

_computation, doi:10.3390/computation8040096_

Round 1
Reviewer 1 Report
In this paper, roughly speaking, the authors proposed a new control operation, so-called, fixed point iteration-based adaptive controller’s operation that is investigated via numerical simulations. Regarding the Lyapunov function-based adaptive control design, this approach is mathematically quite
simple alternative.
The Concept of Using of Fixed Point Iteration-based Control Method is a good example of the application of the fixed point theory that has been studied several researchers. Such connections help also these researchers to improve the observed results.
I recommend this nice paper after all typos are removed. For example,
"fixed point iteration-based adaptive controller’s
207 operation was investigated via numerical simulations."
should be "fixed point iteration-based adaptive controller’s
207 operation is investigated via numerical simulations."
Author Response
Respected Reviewer!
Thank you so much for your time and your valuable suggestions. The details are attached in PDF form.

Reviewer 2 Report
Kindly see the attached file.

Author Response
Respected Reviewer!
Thank you so much for your valuable time and good suggestions to improve the manuscript. The changes that were done are given in the attached PDF file. Please find the attachment.
